# Unraveling the Molecular Mechanism of Traditional Chinese Medicine: Formulas Against Acute Airway Viral Infections as Examples

**DOI:** 10.3390/molecules24193505

**Published:** 2019-09-27

**Authors:** Yi Shin Eng, Chien Hsing Lee, Wei Chang Lee, Ching Chun Huang, Jung San Chang

**Affiliations:** 1Department of Traditional Chinese Medicine, Kaohsiung Medical University Hospital, Kaohsiung Medical University, Kaohsiung 80708, Taiwan; gaziphiphi@gmail.com (Y.S.E.); r98423023@gmail.com (C.C.H.); 2Department of Pharmacology, Graduate Institute of Medicine, College of Medicine, Kaohsiung Medical University, Kaohsiung 80708, Taiwan; 3Department of Medical Research, Kaohsiung Medical University Hospital, Kaohsiung Medical University, Kaohsiung 80708, Taiwan; 4Graduate Institute of Clinical Medicine, College of Medicine, Kaohsiung Medical University, Kaohsiung 0708, Taiwan; weichanglee7202@gmail.com; 5Division of Gastroenterology, Department of Internal Medicine, Kaohsiung Medical University Hospital, Kaohsiung Medical University, Kaohsiung 80708, Taiwan; 6Department of Renal Care, College of Medicine, Kaohsiung Medical University, 100 Shih-Chuan 1st Road, Kaohsiung 80708, Taiwan

**Keywords:** alternative medicine, complementary medicine, prescription, therapy

## Abstract

Herbal medicine, including traditional Chinese medicine (TCM), is widely used worldwide. Herbs and TCM formulas contain numerous active molecules. Basically, they are a kind of cocktail therapy. Herb-drug, herb-food, herb-herb, herb-microbiome, and herb-disease interactions are complex. There is potential for both benefit and harm, so only after understanding more of their mechanisms and clinical effects can herbal medicine and TCM be helpful to users. Many pharmacologic studies have been performed to unravel the molecular mechanisms; however, basic and clinical studies of good validity are still not enough to translate experimental results into clinical understanding and to provide tough evidence for better use of herbal medicines. There are still issues regarding the conflicting pharmacologic effects, pharmacokinetics, drug interactions, adverse and clinical effects of herbal medicine and TCM. Understanding study validation, pharmacologic effects, drug interactions, indications and clinical effects, adverse effects and limitations, can all help clinicians in providing adequate suggestions to patients. At present, it would be better to use herbs and TCM formulas according to their traditional indications matching the disease pathophysiology and their molecular mechanisms. To unravel the molecular mechanisms and understand the benefits and harms of herbal medicine and TCM, there is still much work to be done.

## 1. Significance of Herbal Medicine and Traditional Chinese Medicine (TCM) 

All major cultures have used medicinal plants for health purposes to form herbal medicines over thousands of years. Herbal medicine has been referred to as phytotherapy in some European countries, as well as being a part of ayurvedic medicine in India and Chinese traditional medicine (TCM) in China. Herbs have often been used as first-line therapy for discomforting symptoms commonly found in daily life. To have better effects, several active molecules of herbs had been identified and purified, such as atropine from Belladona (*Atropa belladonna*), digoxin from Foxglove (*Digitalis purpurea*), salicylic acid from Willow bark (*Salix purpurea*), taxol from Pacific yew (*Taxus brevifolia*), and vincristine from Madagascar periwinkle (*Catharanthus roseus*), etc. During the twentieth century, production of synthetic drugs outweighed herbal medicine and largely replaced herbs by their better pharmacologic effects and greater profitability; however, herbal medicine using natural products to improve health care had a resurgence and became popular again in the mid-twentieth century [1]. In 1997 national survey in the USA, 12.1% of adults had used herbs for health care in the previous year; and almost five-fold increase [2]. In 2012 National Health Interview Survey (NHIS) Alternative Medicine Supplement, use of natural products had grown to 17.7% adults in the previous year [3]. Among them, products of medicinal plants were the most popular, including Echinacea, Ginseng, herb pills, ginkgo, etc. [4]. Aside from the USA, 65% of Germans used herbs [5]. In China, people commonly use TCM herbs as dietary supplements. In developing countries, herbs are often less expensive, more accessible, and more popular than orthodox western medicine [6]. With such popularity, it is common to have herbal therapy concomitantly being taken with orthodox western medicine therapy, intentionally or accidentally. Herbs and TCM formulas (i.e., combination of several herbs) contain numerous active molecules, with each active molecule possessing unique pharmacologic activities. Basically, they are a kind of cocktail therapy. Pharmacological activities of herbs may or may not differ from those of their active molecules, but it is common that pharmacological activities of a TCM formula might differ from those of its ingredients, so there might probably be complex herb-drug, herb-food, herb-herb, herb-microbiome, and herb-disease interactions. 

Herbal medicine, including TCM, has the potential for both benefit and harm, might largely affect the health of the population in the world, and has become an important issue, so only after understanding their molecular mechanisms and clinical effects, can herbal medicine and TCM be helpful to human health. Numerous basic and clinical studies of TCM have been carried out in the past, although several important issues remain. Knowing how and why TCM formulas are used, could help us to link molecular mechanisms with their clinical effects. In the current review, we accordingly focus on the current knowledge and shortage of the molecular machismos of TCM formulas using management of acute airway viral infections as experimental example.

## 2. Indications of Use of TCM Formulas for Diseases Management

It is common to initiate the therapy of orthodox western medicine by fitting the pharmacologic characteristics of a drug, including pharmacokinetic and pharmacodynamic effects, to the disease pathophysiology. Physicians further validate the clinical application according to evidence-based medicine (EBM); however, this is not the case for TCM. TCM developed in ancient China, and at that time, physicians managed diseases with herbs only by clinical experience, without any knowledge of disease pathophysiology, nor the pharmacologic activities of herbs, to say nothing of the molecular mechanisms of herbs. To unravel the molecular mechanism of TCM formulas, it would be better to understand how physicians prescribe them first. TCM includes herbal therapy, acupuncture, massage, and dietary therapy. In the current work, TCM will be simply defined as herbal and dietary therapies. TCM is widely popular in East Asia and forms the Kampo medicine in Japan, as well as traditional Korean medicine; importantly, traditional medicines form the mainstream of healthcare in these countries. In ancient China, several famous TCM textbooks summarized the clinical experiences of using TCM formulas against various diseases, including endemic diseases, and each formula has its indication, including specific symptoms of patients. This is quite different from using TCM formulas based on the Yin-Yang theory (two opposite, but complementary forces) and Five Elements theories (everything in the world can be classified into the natural five elements. These elements promote as well as feedback each other to keep everything in balance). Kampo medicine in Japan classified TCM theories into ancient formula sect and recent formula sect for prescribing TCM formulas. The physicians of the ancient formula sect (A-physicians) use the indications of formula formed before the Mongol dynasty of China (1279–1368 A.D.), which are mainly based on clinical experiences. By contrast, the physicians of the recent formula sect (R-physicians) use the Yin-Yang and Five Elements theories, popular after the Mongol dynasty of China, to prescribe TCM formulas [7].

### 2.1. Indications of R-Physicians’ Formulas 

R-physicians propose that diseases develop from disharmony between Yin and Yang or imbalance between Five Elements. To recovery from a disease state, R-physicians largely focus on establishing and maintaining the harmony between Yin and Yang and balance between Five Elements to achieve body homeostasis. R-physicians’ formulas do not focus on the disease or problems of an organ; rather, they try to turn the systematic disharmony of a patient into harmony. By interpreting all patients’ symptoms as problems of disharmony, R-physicians use several ingredients at a time to target the problem and develop a formula. As the interpretation of dis-harmony may change, composition of the formula can also be changed quickly in order to regain the harmony between Yin and Yang and balance between Five Elements. This way of prescribing a TCM formula is rather difficult for physicians of orthodox medicine to understand. Traditionally, R-physicians’ formulas are commonly used to deal with chronic diseases.

### 2.2. Indications of A-Physicians’ Formulas 

A-physicians focus on the indication of a preformed TCM formula in medicinal textbooks. When a patient’s symptoms fulfill the indicated symptoms of a specific TCM formula, A-physicians will prescribe that particular formula. They start with one indicated formula, then add on or delete ingredients according to the symptoms, therefore the composition is relatively unchanged. By the fact that the common diseases in ancient China are infectious, TCM formulas of A-physicians are hypothesized to be able to deal with both acute endemic diseases and chronic diseases. Nevertheless, clinical experiences of the past do not support management of infectious diseases. Antibiotics, but not A-physicians’ formulas, can easily control acute endemic bacterial infection; therefore, without clear effect, TCM then became neglected in China. This viewpoint was changed only after the discovery of the activity against virus of glycyrrhizic acid from *Glycyrrhiza glabra* [8]. From then on, TCM formulas and their ingredients have been hypothesized to be effective against certain viral infections at different stages, and indeed, several active molecules isolated from herbs show antiviral activities, such as cimicifugin from *Cimicifuga foetida* against the Human Respiratory Syncytial Virus [9], and pterodontic acid from *Laggera pterodonta* against the influenza A virus [10], etc.

With the knowledge of symptomatic pathophysiology proposed by orthodox medicine, it is easier to understand the activities and clinical effects of A-physicians’ formulas, as well as to unravel their molecular mechanisms. By contrast, the R-physicians’ formulas are highly variable, from patient to patient, case by case, and from time to time, for a particular disease. It is difficult to interpret their pharmacologic activities by the pathophysiology of symptoms in accordance with the disharmony of patients for a disease. It is also rather difficult to study their pharmacologic activities and clinical effects with different combinations of ingredients, to say nothing of unraveling their molecular mechanisms; accordingly, the current work will focus mainly on the formulas of A-physicians. 

## 3. Factors and Mechanisms Affecting Clinical Effects and Side Effects of the Formulas of A-Physicians

Such formulas are fixed combinations of several herbal ingredients with various pharmacologically active molecules, and these active molecules may act independently or interactively. In this way, they form a kind of cocktail therapy under the expense of unpredictable herb-drug, herb-food, herb-herb, herb-microbiome, and herb-disease interactions. Complex interplay between active molecules of TCM formulas, disease pathophysiology, and individual gene-based metabolism, makes the therapeutic responses unpredictable. Unpredictable herb-drug interactions may be further worsened when TCM therapy is combined with therapeutic agents of orthodox medicine. Active ingredients of a TCM formula can have active molecules that are pharmacologically different from that ingredient or the TCM formula. Obversely, the pharmacological activities of a TCM formula may differ from those of their active ingredient or active molecules of ingredients. As a consequence, unraveling the molecular mechanism of a TCM formula needs comparison of the pharmacological activities between the formula, its ingredients, and the active molecules in the ingredients.

The amount of most bioactive compounds in the herbs is very low. Combination of herbs to form a TCM formula can further decrease their concentrations. Is it possible that herbs and TCM formulas can be effective in this low concentration of bioactive compounds? Is it possible that little amount of bioactive molecules can cause interactions? In orthodox western medicine, vitamins of little amount can show their clinical effects. The molecular mechanisms are the key, instead of their amount. In the real world practice, herbs and TCM formulas are bioactive. Several side effects of TCM formulas have been reported [11,12,13,14,15,16,17] that raise the safety issue of TCM formula. Natural products and TCM formulas might not be safe; however, some A-physicians suppose that the side effects may come from the misuse of TCM formulas without fulfilling their indications, while R-physicians consider the side effects developing from the misinterpretation of the disharmony. Most TCM physicians do not agree that these side effects come from TCM formulas themselves, although with the same indications or disharmony, it is common to find that some patients respond well while others do not—while some patients develop side effects, some show responses opposite to the in vitro pharmacological effects. For example, *Panax quinquefolius* prolongs thromboplastin time, prothrombin time (PT) and thrombin time in vitro [18]. However, in combination therapy with warfarin, *Panax quinquefolius* actually decreases seral concentration of warfarin and has shortened INR in a clinical trial [19]. With therapy using *Panax ginseng*, one can develop thrombosis [20], bleed [21,22], or remain without any particular response [23]. 

Several factors may affect the molecular mechanisms and subsequent clinical effects of TCM formulas, including individual gene-based response, composition and amount of active molecules in TCM formulas, complex interactions, and appropriateness of use of TCM formulas. 

### 3.1. Individual Gene Affecting Pharmacodynamics and Pharmacokinetics 

Individual genetic basis is unique to metabolize TCM formulas and produces different responses. From the results of pharmacokinetic study of Gan-Lu-Siao-Du-yin, a TCM formula (submitted data), the blood concentrations of structurally-related index molecules, baicalin and baicalein, wogonin, and wogonoside, are highly variable between participants. The different patterns of blood concentrations support the unique pharmacokinetic profile based on individual genes. Different concentrations of active molecules may affect the pharmacologic activities. Therefore, individual gene-based metabolism could be one of the major factors affecting the molecular mechanisms and subsequent clinical effects of TCM formulas. To provide insights into action mechanisms of TCM formulas, metabolomic technologies might be helpful. A metabolomics integrative approach accepts a ‘top-down’ strategy to express the function of organisms through terminal symptoms of metabolic network and will gain a revolution in understanding of the holistic concept of TCM [24,25]. Such technologies have been used to investigate the biological mechanisms of different syndromes of patients by studying the functional activities of the human body from a system-wide perspective. For example, the overall biological characterization of the urine of psoriasis patients with TCM blood stasis syndrome was performed to investigate the therapeutic metabolomic mechanism of the Optimized Yinxieling formula [26]. In addition, metabolomics have been considered a powerful tool in diagnosis and treatment of primary dysmenorrhea by supporting information on changes of metabolites and changes in endocrinal, neural, and immune pathways [27]. The Xiang-Fu-Si-Wu formula has been demonstrated to affect some significant perturbations in sphingolipid and glycerophospholipid metabolism as well as steroid hormone biosynthesis to make the metabolic discrepancy return to the normal level [28]. 

### 3.2. Complex Active Molecules vary in TCM Formulas

TCM formulas are complex with numerous active chemical molecules in variable amounts. Among these molecules with different pharmacologic activities, it is unclear which one mainly accounts for the clinical effect of a TCM formula, as the most abundant molecule might not be the most important one for a specific activity. The amount of an active molecule can be easily changed in different batches of product, or by different agriculture and collection of medicinal plants, therefore, understanding the molecular mechanisms of a TCM formula requires analysis not only of the mechanisms of the TCM formula as a whole, but those of individual active molecules and ingredient respectively as well. Understanding the molecular mechanisms of an active molecule can facilitate its development into an investigational new drug (IND). Meanwhile, unraveling the molecular mechanisms of a TCM formula can help to validate its traditional use and avoid its misuse and side effects. To keep a relatively constant amount of active molecules and pharmacologic activities, several things should be paid attention to, including use of right specie, use of right part of a plant, and use of a plant harvested in the right season. Both use of closely related but wrong species and use of wrong part of herbs might lead to different active molecules with different pharmacologic activities, and various clinical effects and side effects. Plants harvested in different seasons might contain variable amounts of active molecules thereby affecting their activities. Some active molecules are secondary metabolites of plants against physical, chemical, or biological stimulants. Active molecules of some plants can vary from year to year and place to place. Therefore, confirmation of its authenticity is the cornerstone. 

In addition to this, fingerprints of the active molecules are needed to confirm the authenticity of a plant or a formula and to confirm the amount of active molecules via high-performance liquid chromatography (HPLC) or liquid chromatography coupled with mass spectrometry (LCMS). This is highly necessary for quality control and efficacy assessment. To have quality control of the products of TCM formulas, good manufacturing practice (GMP) procedure should be followed to avoid (a) inadequate processing that might lead to different chemical compositions of the final product; (b) inadequate storage conditions or prolonged storage that might lead to microbial contamination and early decay of the active molecules; and (c) adulteration of formulas and accidental contamination creating serious uncertainty in quality. Adulteration is a plant or formula containing active pharmaceuticals or other bioactive agents for the purpose of claiming better efficacy or broader indications. Accidental contamination is the plant raw materials containing heavy metals or other toxic substances from the manufacturing process due to ecological collapse. Lead, mercury, and arsenic contamination in traditional Chinese herbs has been reported [29,30,31,32] 

### 3.3. Complex Interplays between Herb-Drug, Herb-Food, Herb-Herb, Herb-Microbiome, and Herb-Disease

In the clinical practice of orthodox medicine, the more drugs used, the more adverse drug reaction (ADR) occurred [33]. This is commonly caused by drug-drug interactions. TCM formulas are mixtures of several ingredients. Each ingredient has several bioactive compounds, so a TCM formula has numerous bioactive compounds. Thinking of dozens or even hundreds of active molecules in a TCM formula been taken at once implies that the probability of drug interaction could be high. Such interactions may be found between herb and drug, herb and food, herb and herb, herb and microbiome, and even between herb and disease. For example, in herb-drug interaction, *Scutellariae baicalensis* is a common ingredient in TCM formulas. *S. baicalensis* contains baicalin, a flavonoid, as one of its major molecules. Interactions between *S. baicalensis* and drugs are found due to baicalin affecting metabolic enzymes of drugs, displacing plasma protein binding, and regulating various transporters involved in the pharmacokinetics [34]. Baicalin may inhibit the expression and hydroxylation activity of CYP3A in the liver to change the pharmacokinetics of drugs [35]. Co-administration of extract of *S. baicalensis*, and mefenamic acid, a kind of NSAID, can potentiate its anti-inflammatory effect [36]. Co-administration of baicalin and rosuvastatin, a HMG-CoA reductase inhibitor commonly used to reduce serum cholesterol level, might find reduced plasma concentration of rosuvastatin in certain patients with certain genomes [37]. As for herb-food and herb-food-drug interactions, baicalin can potentiate the antioxidant activity of β-carotene, which is a terpenoid of red-orange color, abundant in plants and fruits [38]. The intakes of flavonoid-rich foods and beverages, containing baicalin and rutin, might compete with the binding site of calcium channel blockers on human serum albumin to affect their clinical effects [39,40]. By contrast, baicalin and rutin will increase the binding affinity of curcumin on human serum albumin to change its bioavailability [41]. Herbs may also interact with each other. For example, baicalin and berberine are important coexisting molecules of the combination of *S. baicalensis* and *Coptidis chinensis*. Berberine, but not baicalin, can increase glucose consumption. Co-administration of berberine and baicalin had a synergetic effect on glucose utilization [42]. Additionally, TCM herbs are commonly used as food supplements and dietary therapy. Foods have been reported to modify the intestinal microbiome [43]. Commensal microbiota have been thought to be involved in the development of the innate and adaptive immunity, nutrient metabolism of humans, and protection from the overgrowth of intestinal pathogens [44]. Herbs can change pharmacokinetics of drugs by intestinal microbiota [34]. The intestinal microbiome is metabolically active to play an important role in the absorption of certain active molecules and change their bio-availabilities, particularly in those containing glycosidic linkages [34,45]. Therefore, change of intestinal microbiome by herbs-containing foods may affect human health care and drug therapy. As for interactions between disease and herb, more absorption of active molecules of Maxing Shigan decoction (MXGST), including liquiritin, glycyrrhizin, amygdalin, prunasin, ephedrine, pseudoephedrine, and methylephedrine, can be found in RSV pneumonia-infected rats vis-à-vis normal rats by reducing the clearance rates of these active molecules [46]. There are highly complex interactions between herbs and drugs, foods, herbs, microbiome, and diseases, and most of these complex interactions are not completely discovered or remain unseen, just like the submerged part of an iceberg. Therefore, there are still insufficient data to completely understand the molecular mechanisms, pharmacokinetics, pharmacodynamics, and interactions of TCM formulas. 

## 4. Unraveling the Molecular Mechanism of TCM Formulas, Using Airway Viral Infections as Examples

Acute airway infections, including acute bronchitis, viral pneumonia, and acute exacerbation of chronic obstructive pulmonary disease (COPD), are commonly caused by viruses of different families, including rhinovirus, influenza, and parainfluenza virus, enteroviruses, coronavirus, adenovirus, respiratory syncytial virus, etc. [47,48]. These viruses infect epithelia, produce inflammation, induce immune response, and cause symptoms. From the viewpoint of pathophysiology, TCM formulas used to manage airway viral infections need to have antiviral activity against such viruses listed above, and/or to induce antiviral cytokines, and/or anti-inflammatory effect, and/or to relieve symptoms commonly presented in airway infections (Figure 1). To simplify the molecular mechanisms and to correlate the pharmacologic activities with their clinical effects, five formulas of A-physicians will be used as examples against airway infections:

### 4.1. Ge-Gen-Tang (GGT; Kakkon-To in Japan; Galgeun-Tang in Korea; Gegen Decoction in China)

Ge-Gen-Tang (GGT; Table 1) [49] has been reported to be effective in the treatment of common colds, chronic sinusitis, allergic rhinitis, and pneumonia. The indication to use GGT is patients with symptoms of headache, fever without sweating, and particularly stiffness of neck and shoulders. GGT can successfully reduce various symptoms of dogs infected with common cold viruses [50]. GGT has antiviral activities against respiratory syncytial virus (RSV) [51] and influenza virus [52]. GGT can reduce the mortality of influenza virus-infected mice [53]. Among its active molecules, uralsaponins from *Glycyrrhiza uralensis* [54] and procyanidin from *Cinnamomum cassia* [55] may effectively inhibit the replication of the influenza virus. Allicin in ginger (*Zingiber officinale*) and coumarin in *Cinnamomum cassia* might have the activity to inhibit influenza neuraminidase [56]. Paeonol and 1,2,3,4,6-penta-O-galloyl-β-d-glucopyranose from *Paeonia lactiflora* show antiviral activity against rhinovirus [57]. Of its anti-inflammatory effects, GGT can suppress the interleukin-1α (IL-1α) production induced by interferons (IFN) in influenza [58]. GGT was found to decrease cigarette smoking- (CS-) and lipopolysaccharide (LPS)-induced elevated counts of inflammatory cells, and expression of inflammatory cytokines and proteins (IL-6, TNF-α, iNOS, and COX-2) [59]. GGT can stimulate IL-12 and IFN-β to counteract viral infection [53], and can also enhance the phagocytic activity of macrophages [50]. Among its active molecules, paeoniflorin, a major constituent of *Paeonia lactiflora*, exerts anti-inflammatory and immunomodulatory effects by balancing the function of Th1/Th2 [60]. Glycyrrhizin, a major constituent of *Glycyrrhiza uralensis*, suppresses nuclear factor-kappa B (NF-κB) via the phosphoinositide 3-kinase (PI3K) pathway, inhibits the production of nitric oxides (NO), prostaglandin E2 (PGE2), and reactive oxygen species (ROS), and reduces the protein and mRNA levels of inducible NO synthase (iNOS) and cyclooxygenase-2 (COX-2) [61](Figure 2). Meanwhile, glycyrrhiza polysaccharide, isolated from *Glycyrrhiza uralensis*, significantly induces NO production and iNOS transcription in peritoneal macrophages [62]. However, this study design uses intraperitoneal injection to show the induction of NO [62], instead of the traditional oral route. Polysaccharides will be normally digested in the gastrointestinal tract into monosaccharide, so that polysaccharides can hardly reach intraperitoneal macrophages in the regular oral route, so this pharmacologic activity has been questioned. Isoliquiritigenin, a flavonoid from *Glycyrrhiza uralensis*, inhibits NF-κB activation to suppress inflammatory response [63]. Cinnamaldehyde, from *Cinnamomum cassia,* inhibits the secretion of PGE2, IL-1β and tumor necrosis factor-α (TNF-α), and the activation of NF-κB to show the anti-inflammatory effect [64,65,66]. Particularly, e-cinnamaldehyde and o-methoxy-cinnamaldehyde down-regulate NO and TNF-α production to show anti-inflammatory activity [67]. Although it can mediate antiviral activity, TNF-α plays only a minor role in clearance of various airway viruses; rather, it is the major contributor of T-cell-mediated lung injury [68]. For enhancement of antiviral immunity, puerarin, an isoflavonoid from *Pueraria lobate*, increased IFN-γ [69]. 6-Gingerol (6-G), the main bioactive component of ginger (*Zingiber officinale*), increases IFN-γ and IL-12, but decreases IL-10 and transforming growth factor-β1 (TGF-β1) levels [70]. On the contrary, gingerol was also reported to suppress T cell response and inhibit IFN-γ synthesis [71](Figure 2). These conflicting data may come from different study designs and raise questions about the actual pharmacological activity in humans.

### 4.2. Ma-Huang-Tang (MHT; Maoto in Japan; Mahuang Decoction in China)

Ma-Huang-Tang (MHT; Table 2) [49] have been reported to be effective in the treatment of Influenza, upper respiratory tract infection, acute and chronic bronchitis, and asthma [72,73]. The indication to use MHT is patients with chills, fever without sweating, headache, shortness of breath, and joint pain. Clinically, MHT can effectively reduce fever and flu symptoms, including myalgia, headache, arthralgia, fatigue and cough, in patients with seasonal influenza type A [74]. Among its active molecules, L-ephedrine from *Ephedra sinica* and amygdalin from *Prunus Armeniacae,* possess the antitussive effect [75], so co-administration of *Ephedra sinica* and *Prunus Armeniacae* shows a better antitussive effect than use singly [76]. MHT clearly shows anti-inflammatory activity via suppressing the NO/PGE2 pathway [77], reducing inflammatory cells infiltration and reducing pro-inflammatory cytokine, including TNF-α, IL-1β, and IL-6, in lung experiments [78]. In an acute bronchial asthma mice model, MHT can also mitigate the pathological changes of acute asthma-like syndrome through inhibition of the Toll-like receptor 9 (TLR9) pathway [79]. MHT can modulate Th1/Th2 cytokines via decreasing IL-4 & IL-17 and increasing IFN-γ levels. MHT can inhibit TH17 cells [80], and decreases IL-4, IL-5, TNF-α, CD3+, CD8+ T cell levels (Th2 response), but increases IL-2, IFN-γ, and CD4+ T cell levels (Th1 response) to increase CD4+/CD8+ ratio [81]. 

Among its active ingredients, *Ephedra sinica* inhibits PGE2 biosynthesis, reduces IgE-mediated histamine release, reduces the mRNA or protein levels of IL-1β, IL-6, TNF-α, COX2, and NF-κB [82] and inhibits complement activation of both classical and alternative pathways [83]. Additionally, *Ephedra sinica* can directly activate both alpha- and beta-adrenergic receptors to reduce bronchial mucosal edema and to dilate the bronchus respectively [75,84]. To understand the molecular mechanism, several active molecules have been identified. Ephedrannin A and B, from *Ephedra sinica*, effectively suppressed the transcription of TNF-α, IL-1β, and NF-κB, and the phosphorylation of p38 mitogen-activated protein (MAP) kinase to exert their anti-inflammatory actions on LPS-stimulated macrophages [85]. *Glycyrrhiza uralensis* and *Cinnamomum cassia* contain active molecules mediating anti-inflammatory and immunomodulatory effects mentioned in the GGT section. For its antiviral activity, MHT was initially thought to inhibit airway viruses through inducing antiviral IFN. However, herbacetin from ephedrine alkaloid-free extract of *Ephedra sinica* might have anti-influenza activity similar to its extract containing ephedrine and pseudoephedrine [86]. The study of *Ephedra sinica* is relative rare owing to it containing ephedrine and pseudoephedrine, which are illegal in several countries. *Glycyrrhiza uralensis* and *Cinnamomum cassia* also have active antiviral constituents mentioned in the GGT section (Figure 3). 

### 4.3. Ma-Xing-Gan-Shi-Tang (MXGST; Maxing Shigan decoction in China)

Ma-Xing-Gan-Shi-Tang (MXGST; Table 3) [49], a similar formula to Ma-Huang-Tang, is effective against Influenza virus infection [87]. MXGST has only one ingredient different from that of MHT, i.e., using gypsum instead of *Cinnamomum cassia*, so their pharmacologic effects and active molecules are similar, except that gypsum possesses a more powerful anti-pyretic effect by decreasing the PGE2 level in the hypothalamus [88]. Co-treatment with *Ephedra sinica* and gypsum can have synergistic effects to manage fever and asthma than single use [89]. MXGST add-on therapy may improve pulmonary function indicies, such as forced expiratory volume in one second (FEV1), forced vital capacity (FVC), and FEV1/FVC in patients with acute exacerbation of COPD [90]. Additionally, at higher cumulative doses, MXGST might reduce the incidence of pneumonia and protect against admission [91]. The indication to use MXGST is patients with fever, cough with yellow and sticky sputum, chest pain, and shortness of breath. MXGST has been found to have antitussive and anti-pyretic effects in an LPS-induced hyperthermia rat model [92]. MXGST has bronchodilation effect mediated by stimulation of β2-adrenoceptors in pigs [93] and can block acetyl-cholinergic and histaminergic receptor-induced bronchial contraction in rats [92], reduce neutrophilic inflammation [93], and in a COPD rat model, can decrease IL-4, IL-8, and TNF-α, but increase IFN-γ [94]. These effects may be beneficial to manage airway viral infections with cough. The molecular mechanism of MXGST is summarized (Figure 4).

### 4.4. Xiao-Qing-Long-Tang (XQLT; Sho-Seiryu-To in Japan; So-Cheong-Ryong-Tang in Korea; Minor Blue-Green Dragon Decoction in China) 

Xiao-Qing-Long-Tang (XQLT; Table 4) [49] is one of the most common prescriptions used against allergic rhinitis [95]. XQLT, at higher cumulative doses, might reduce the incidence of pneumonia and protect against admission [91]. XQLT with/without Ma-Xing-Gan-Shi-Tang (MXGST) is the most frequently prescribed TCM formula for COPD [96], and is also commonly used in the treatment of patients with respiratory diseases, such as common cold, flu, bronchitis, asthma, bronchiectasis, and emphysema. The indication to use XQLT is patients with cough, watery rhinorrhea or watery sputum, but without thirst. To manage airway viral infection, XQLT is effective against human respiratory syncytial virus (RSV) infection by preventing viral attachment, internalization, syncytial formation, and by stimulating IFN-β secretion [97], and has been proven beneficial against influenza virus in vivo through the augmentation of antiviral IgA antibody [98,99]. XQLT can reduce the airway inflammation with the decrease of eosinophils count, the ovalbumin (OVA)-specific immunoglobulin E (IgE) antibody, and histamine release [100,101,102], can also modulate Th1/Th2 balance thereby reducing IL-4 and restoring IFN-γ levels [101,103]. Among its active molecules, schisandrin A, a bioactive lignin of *Schisandra sphenanthera,* inhibits the IL1β-induced inflammation via suppression of mitogen-activated protein kinase (MAPK) and NF-κB signal pathways [104], and can also inhibit the NF-κB, MAPK and PI3K/Akt pathways, partially mediated by the activation of the Nuclear factor erythroid 2-related factor 2/Heme oxygenase-1 (Nrf2/HO-1) pathway to manage inflammatory and oxidative disorders caused by over-activation of macrophages [105]. Schisandrin B, another bioactive lignin of *Schisandra sphenanthera,* increases the expression of Nrf2 and HO-1 and blocks the activation of NF-κB induced by LPS to suppress the production of vascular cell adhesion molecule 1 (VCAM-1), intercellular adhesion molecule 1 (ICAM-1), TNF-α, and IL-8 expressions in human umbilical vein endothelial cells (HUVECs) [106]. α-Cubebenoate, isolated from *Schisandra chinensis*, can block the increase of IL-1β and IL-6 during inflammation [107], and inhibit LPS-induced expression of iNOS and COX-2 [108]. Oral polysaccharide from *Schisandra chinensis* showed antitussive effect in a guinea pig model [109]. The active molecules of common ingredients, including *Ephedra sinica*, *Cinnamomum cassia*, *Paeonia lactiflora*, *Glycyrrhiza uralensis*, and *zingiber officinale*, have several pharmacologic activities mentioned in the above sections. Most of these activities aim at inhibiting inflammation induced by airway infection, however, lectin from *Pinellia ternate* may activate macrophages, induce neutrophil migration, cytokine release, ROS overproduction and the activation of the NF-κB signaling pathway to produce pro-inflammatory activity [110](Figure 5). Therefore, interactions, including both synergistic and antagonistic between active molecules in a TCM formula could be so complex as to affect the final effects.

### 4.5. Ye-Gan-Ma-Huang-Tang (YGMHT; Yakammaoto in Japan; Shegan-Mahuang-Tang or Sheganmahuang Decoction in Chinese)

From more than two thousand years ago in China and Japan, Ye-Gan-Ma-Huang-Tang (YGMHT; Table 5) [49] has been used to manage flu-like symptoms. A meta-analysis shows that YGMHT can improve the total effective rate, FEV1, and asthma control test (ACT) score of refractory asthmatic patients [111]. When combined with salbutamol aerosol, YGMHT can obviously improve their pulmonary functions, including ACT score, PEF, FEV1, and FEV1% predicted value. The indication to use YGMHT is patients with chill and fever, hyperinflation of lung, cough with stridor and rales associated with frothy or whitish sputum [112]. YGMHT has been reported effective against enterovirus infection, including coxsackievirus [113] and EV71 [114]. It can regulate the serum levels of TNF-α, IL-10, and IL-13 to show clinical effect in management of cough and variant asthmatic symptoms in children [115]. Additionally, a modified YGMHT (also named San-Long-Ping-Chuan-Decoction; SLPCD) can significantly inhibit airway inflammation, reduce inflammatory cells in bronchoalveolar lavage fluid (BALF), and decrease the serum total IgE levels [116]. SLPCD can significantly down-regulate the mRNA expression levels of Th2 cytokines (IL-4, IL-5, IL-10, and IL-13) and up-regulate those of Th1 cytokines (IL-2 and IFN-γ) in lungs of asthmatic mice [116]. For this anti-inflammatory activity, *Aster tataricus* can protect from LPS-induced acute lung injury mainly through inhibiting the release of inflammatory cells (WBC, macrophage, neutrophil, lymphocyte), regulating the pro-inflammatory cytokines (IL-1β, IL-6, TNF-α), and attenuating the pulmonary edema [117]. Among its active molecules, *irigenin,* a major active constituent of *Belamcanda chinensis*, can reduce NO and PGE2 production by decreasing the mRNA and protein expression of iNOS and COX-2, respectively, as well as by suppressing NF-κB activation [118](Figure 6). 

To unravel the molecular mechanism of TCM formulas, several issues need to be solved. Many pharmacologic activities are obtained from in vitro and animal studies. Could these activities be extrapolated into humans? There are so many active molecules with various pharmacologic activities in a TCM formula or herbs. Are these molecules specific for that specific activity of a TCM formula or herbs? Could they reach a minimal serum level to exert that particular pharmacologic activity in humans? Active molecules may have the same activity with different potency. The most abundant one may not be the most important one for a particular activity. Is it possible that there might be unidentified active molecules that actually account for a particular pharmacologic activity? Several pharmacologic activities of a TCM formula cannot find a corresponding active molecule. Is it possible that interactions between active molecules, e.g., synergism, but not active molecules themselves, account for a specific activity? Is it possible that new active molecules are generated during preparation of the TCM formula? 

The genetic basis will affect the drug pharmacokinetics. Is there a specific gene or single nucleotide polymorphism (SNP) largely affecting the pharmacokinetic profile? Does publication bias exist so that undesired pharmacologic effects are not published? Is it possible that a particular gene is prone to a specific ADR of a TCM formula or herbs? Could a specific ADR come from interactions between specific active molecules? Several TCM formulas clearly show that some ingredients are not active in managing their traditional applications. Could these inactive ingredients be omitted? All these questions need many studies to determine valid answers. At this moment, it would be better to use herbs and TCM formulas according to their traditional indications and the disease pathophysiology by matching their molecular mechanisms.

## 5. Limitations of Herbal Medicine and TCM

As the world population has continued to age, the elderly have become associated with chronic diseases requiring multiple medications. Increasing dissatisfaction with orthodox medicine and/or preference for alternative therapists and/or naturopathy has meant people continue to seek alternatives to maintain or improve their health, and this has spurred the growth of complementary and alternative medicine (CAM) therapies. The validity of pharmacologic studies, safety issues, and validation of clinical effects of herbal medicine and TCM are limitations that should be highly considered; however, most clinicians are not familiar with herbal medicine and TCM so they do not recommend herbal therapies. More effort should be placed on unraveling the molecular mechanisms in order to solve the above issues. The reasons that limit clinicians from becoming more familiar with herbal medicine and for not recommending herbal therapies are explored below.

### 5.1. Away from Medical Education of Orthodox Medicine

Herbal medicine and TCM form a part of complementary and alternative medicine (CAM). However, evidence-based research in the field of CAM therapies is still limited. There is a wrong perception that a naturally derived product is relatively safe. It is highly important to identify both usefulness and safety of CAM and integrate these health approaches with orthodox medicine through rigorous scientific investigation to improve health care. It is also important for medical educators and providers to recognize the trend, the evidence, the benefits, and the risks of herbal medicine and TCM to educate clinicians for appropriate patient management and education. 

TCM studies published in Chinese are usually not translated into English. The terminology of TCM is also difficult to translate, particularly those used by R-physicians, so medical education of herbal medicine and TCM has been neglected worldwide, except in Germany and China. With their increasing use since the 1960s, understanding the molecular mechanisms, benefits, and limitations by physicians has become increasingly important to monitor their benefits and harmfulness.

### 5.2. Lack of Tough Evidence of Clinical Efficacy 

Most of the evidence supporting clinical claims of herbs and natural products come from studies of inadequate design that do not provide tough evidence of the effects. Relatively few well-designed studies support their clinical claims. Sometimes, adequately powered, double-blinded, placebo-controlled clinical trials come to conclusions against previous reports, such as Echinacea for upper respiratory infection [119], or ginkgo for dementia or mild cognitive impairment [120]. Additionally, the amounts of most bioactive compounds in herbs and TCM formulas are very low. Is it possible that the clinical effects of herbs and TCM formulas can be effective with such a small amount of bioactive compounds? One study discussed the clinical effect of Ma-Huang-Tang (MHT) against seasonal influenza [121]. MHT showed an equivalent clinical effect as neuraminidase inhibitors. However, not every TCM formula can provide such evidence. 

Several TCM formulas, such as Ma-Xing-Gan-Shi-Tang (MXGST), Ge-Gen-Tang (GGT), and Xiao-Qing-Long-Tang (XQLT), are among the top ten most common TCM prescriptions for patients with upper respiratory tract infections (URTIs) in Taiwan [122]. They are commonly used for URTIs, but more research is required to validate their clinical effects and mechanisms. Without tough clinical evidence and clear molecular mechanisms, physicians tend to avoid herbal therapies. Most clinical claims of herbal therapies are based on bench studies that do not possess external validity to support their conclusions; for example, using animal-derived cancer cell lines to study antiviral effects in humans; using cancer cell lines to study physical changes; using intraperitoneal injection to study oral medications; and using high-dose pharmacological designs to study physiological responses, etc. Therefore, there is still much space for improvement of our understanding of the mechanisms of herbal medicine. 

### 5.3. Safety Issue is not Completely Resolved

Herbs are pharmacologically active and can positively or negatively affect patient’s health. With increasing use of herbs as dietary supplements and alternative therapy, there is an increased risk of negative impacts, such as adverse effects and interactions. Herbal medicine is among the most common causes of drug-induced liver injury (DILI) [123]. Additionally, several adverse effects of commonly used herbs have been reviewed, including hypoglycemic or hyperglycemic effect, hypolipidemic or hyperlipidemic effect, hormonal activities, hypertensive, cardioactive [124], and hepatotoxic effects [125]. Some of them are serious even in recommended dosage, such as Ephedra alkaloids (derived from *Ephedra sinica* or Ma huang) [126,127,128]. However, most of the molecular mechanisms causing side effects are unknown. Plants containing pyrrolizidine alkaloids can lead to hepatotoxicity and venoocclusive disease, possibly by the accumulation of toxic metabolites produced via the cytochrome system [129]. Some herbal treatments containing aristolochic acid (AA), including *Aristolochia fangchi*, *Aristolochia debilis* Sieb. et Zucc., *Aristolochia manshuriensis*, *Aristolochia debilis*, can cause AA nephropathy requiring renal replacement therapy [130,131]; Additionally, AA is associated with urothelial cancers [132]. Although most of the above issues have been identified, the unrecognized issues are just the tip of the iceberg.

### 5.4. Complex Interactions are not Fully Understood

TCM formulas have numerous bioactive compounds to form a kind of cocktail therapy. However, the amounts of each bioactive compound in herbs and TCM formulas are very low. This may raise questions about the likelihood of their interactions with others during administration in a decoction. Ginseng, a common natural product used among adults [4], has about 30 ginsenosides as its major bioactive compounds [133], although the actual amount of each ginsenoside is small. After oral administration, ginsenosides are metabolized and transformed by intestinal microbiota. Diet can markedly influence the transformation of ginsenosides. They exert pharmacologic effects in animals, and also show various clinical effects in randomized controlled trials, including several side effects and drug interactions [133], so small amounts of bioactive compounds do have clinical effects and side effects, which might come from the individual pharmacological activity of bioactive compounds or their synergism. The side effects may also come from individual unfavorable bioactivity or from interactions. Additionally, herbal medicine and TCM therapy are commonly used in combination with orthodox medicine by patients, sometimes unknown to their doctors. The complex and unknown interactions, including herb-drug, herb-herb, herb-food, herb-microbiome, and herb-disease interactions, make use of herbal medicine more complicated and requiring frequent monitoring. The mechanisms of these interactions can be divided into molecular mimicry and pharmacologic interactions, such as pharmacodynamic and pharmacokinetic interactions. For example, with molecular mimicry, several herbs naturally has coumarin or salicylate analogue that may potentiate the bleeding risk of warfarin and salicylate, respectively. For pharmacodynamic interactions, *Ephedra sinica* (Ma huang) should not be used with sedative or anti-hypertensive agents. For pharmacokinetic interactions, St John’s wort and several TCM formulas have been noted to affect the CYP450 system of liver, which may increase or decrease the effects of other drugs [134]. Additionally, the herb-drug interaction may be individualized, e.g., in combination with warfarin, *Panax ginseng* may cause thrombotic event [20], bleeding [21,22], or neither [23]. Currently, most of the molecular mechanisms of these identified interactions are not fully understood, to say nothing of the unrecognized interactions.

### 5.5. Quality Uncertainty of Commercially Available Natural Products 

Quality uncertainty impacts the reproducibility of clinical efficacy and safety of commercially available natural products. Variability of the quality of natural products may come from a lack of standardization of manufacturing, including misidentification of authenticity, inadequate processing, inadequate storage, adulteration of formulas, and accidental contamination [135]. Most of those quality problems can be gradually solved by following the WHO guidelines on good agriculture and collection practices for medicinal plants [136] and botanical drug development guidance for industry of FDA [137].

### 5.6. Pitfall of Interpretation of Benefits 

Several health benefits of herbal medicine and TCM are claimed; for example, herbs and TCM formulas, including those discussed above, are believed to have anti-oxidative activities helpful against several diseases. This idea is based on reactive oxygen and nitrosative species (ROS/RNS) as metabolic byproducts that can cause damage to cellular macromolecules; thus, many diseases can be triggered by oxidative stress under high levels of ROS/RNS. These diseases include cancer, inflammation, and degenerative diseases. Oxidative stress causes damage either with an overwhelming production of ROS/RNS or under insufficient levels of antioxidants or repair mechanisms, so blocking the generation of ROS/RNS might prevent and/or manage these diseases [138]. However, ROS/RNS are also signaling molecules for several physiological functions, including regulation of vascular tone, control of ventilation, and erythropoietin production, etc. Actually, ROS-mediated responses may protect against oxidative stress [139]. Additionally, ROS/RNS may play a dual role in different diseases, i.e., ROS/RNS might contribute to, or counteract, the disease progression. It remains unclear that more dosage of antioxidants is not better and may even worsen a medical condition [138]. There are insufficient data to establish the ability of TCM to decrease ROS/RNS levels and establish its effects on the disease, and this affects the interpretation of any claims of benefit. To validate such claims of the benefits of herbal medicine and TCM, much work remains to be done. 

Only when these limitations can be minimized, can the molecular mechanisms of herbs and TCM formulas be understood. Consequently, clinicians can help patients by giving adequate prescriptions and suggestions to minimize the harmfulness and maximize the benefits to healthcare.

## 6. Conclusions 

Herbal medicine, including TCM, is commonly used worldwide. Herbal remedies contain numerous active molecules to form a kind of cocktail therapy. These active molecules may interact with each other to affect the therapeutic effects and produce side effects. Understanding their pharmacological effects, interactions, side effects, clinical effects, and the underlying molecular mechanisms is very important to provide benefits, but avoid harm, to the patient. To unravel the molecular mechanisms, much work remains to be done.

## Figures and Tables

**Figure 1 molecules-24-03505-f001:**
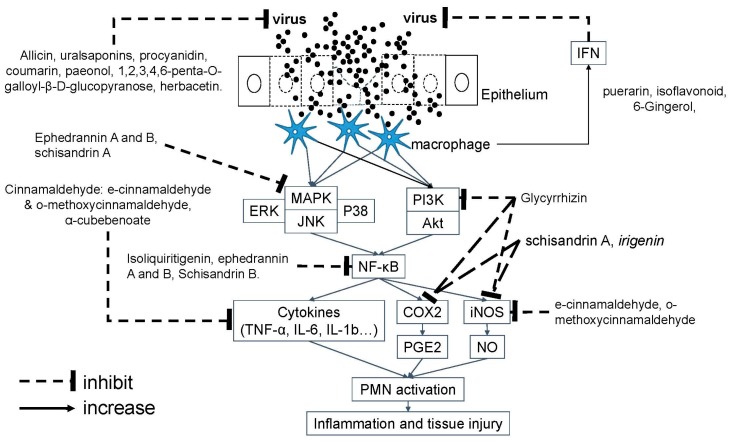
The molecular mechanism of TCM formulas against airway viral infections. Airway viruses infect the epithelium to cause tissue injury. TCM formulas of A-physicians contain several active molecules to inhibit viral replication and signal transduction of inflammatory response. Akt: Serine/threonine protein kinase B (PKB); COX: Cyclooxygenase; ERK: Extracellular signal-regulated kinase; IL: Interleukin; iNOS: Inducible nitric oxide synthases; JNK: c-Jun *N*-terminal kinases; MAPK: Mitogen-activated protein kinase; NF-κB: Nuclear factor-kappa B; NO: Nitric oxide; PG: prostaglandin; PI3K: Phosphoinositide 3-kinase; PMN: Polymorphonuclear neutrophils; TNF: Tumor necrotic factor.

**Figure 2 molecules-24-03505-f002:**
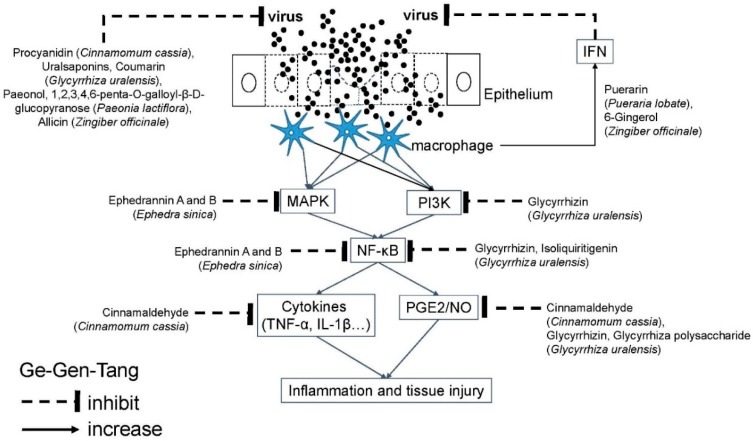
Molecular mechanism of Ge-Gen-Tang against airway viral infections. Airway viruses infect the epithelium to cause tissue injury. Ge-Gen-Tang contains several active molecules to inhibit viral replication and signal transduction of inflammatory response. Akt: Serine/threonine protein kinase B (PKB); COX: Cyclooxygenase; ERK: Extracellular signal-regulated kinase; IL: Interleukin; iNOS: inducible Nitric oxide synthases; JNK: c-Jun N-terminal kinases; MAPK: Mitogen-activated protein kinase; NF-κB: Nuclear factor-kappa B; NO: Nitric oxide; PG: Prostaglandin; PI3K: Phosphoinositide 3-kinase; PMN: Polymorphonuclear neutrophils; TNF: Tumor necrotic factor.

**Figure 3 molecules-24-03505-f003:**
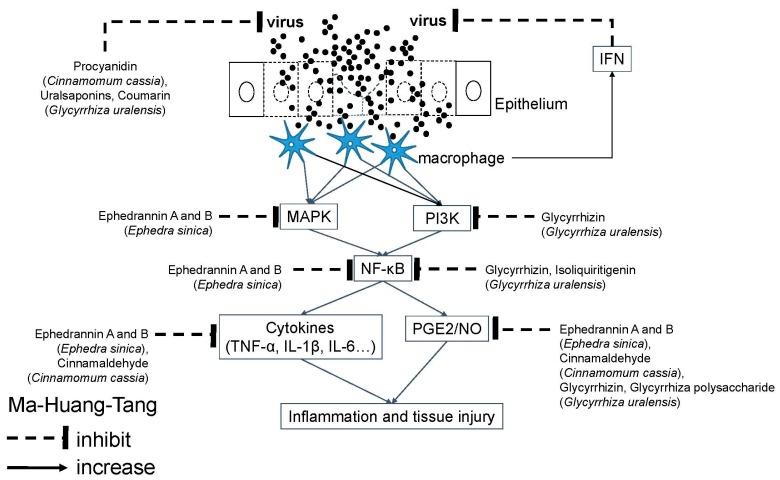
Molecular mechanism of Ma-Huang-Tang against airway viral infections. Airway viruses infect the epithelium to cause tissue injury. Ma-Huang-Tang contains several active molecules to inhibit viral replication and signal transduction of inflammatory response. Akt: Serine/threonine protein kinase B (PKB); COX: Cyclooxygenase; ERK: Extracellular signal-regulated kinase; IL: interleukin; iNOS: Inducible nitric oxide synthases; JNK: c-Jun N-terminal kinases; MAPK: Mitogen-activated protein kinase; NF-κB: Nuclear factor-kappa B; NO: Nitric oxide; PG: Prostaglandin; PI3K: Phosphoinositide 3-kinase; PMN: Polymorphonuclear neutrophils; TNF: Tumor necrotic factor.

**Figure 4 molecules-24-03505-f004:**
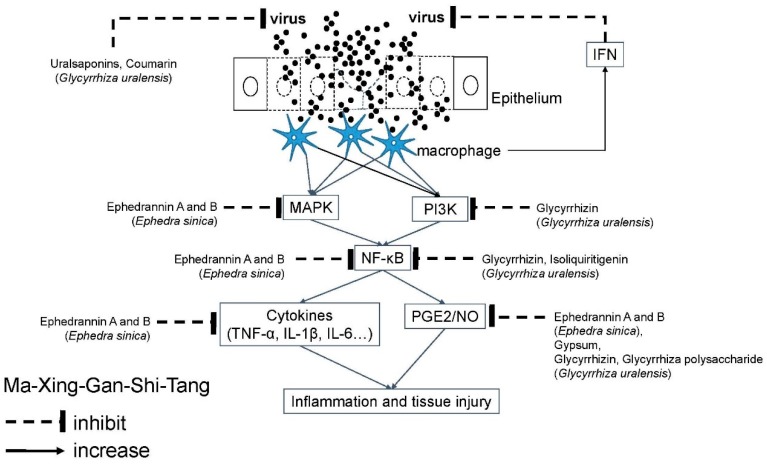
Molecular mechanism of Ma-Xing-Gan-Shi-Tang against airway viral infections. Airway viruses infect the epithelium to cause tissue injury. Ma-Xing-Gan-Shi-Tang contains several active molecules to inhibit viral replication and signal transduction of inflammatory response. Akt: Serine/threonine protein kinase B (PKB); COX: Cyclooxygenase; ERK: Extracellular signal-regulated kinase; IL: Interleukin; iNOS: Inducible nitric oxide synthases; JNK: c-Jun N-terminal kinases; MAPK: Mitogen-activated protein kinase; NF-κB: Nuclear factor-kappa B; NO: Nitric oxide; PG: Prostaglandin; PI3K: Phosphoinositide 3-kinase; PMN: Polymorphonuclear neutrophils; TNF: Tumor necrotic factor.

**Figure 5 molecules-24-03505-f005:**
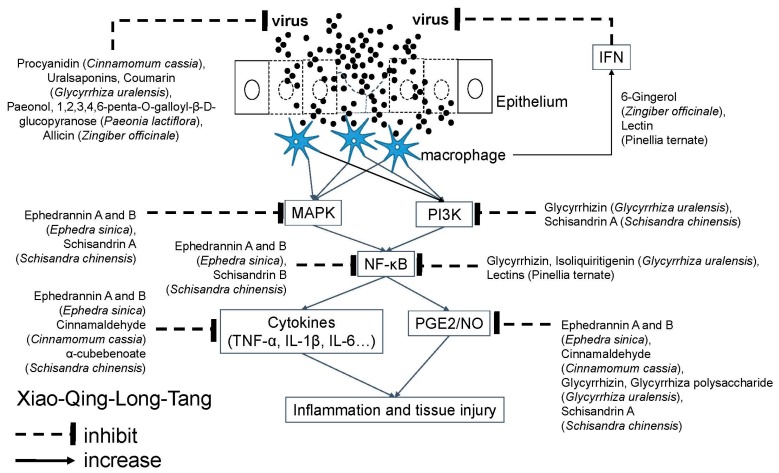
Molecular mechanism of Xiao-Qing-Long-Tang against airway viral infections. Airway viruses infect the epithelium to cause tissue injury. Xiao-Qing-Long-Tang contains several active molecules to inhibit viral replication and signal transduction of inflammatory response. Akt: Serine/threonine protein kinase B (PKB); COX: Cyclooxygenase; ERK: Extracellular signal-regulated kinase; IL: Interleukin; iNOS: Inducible nitric oxide synthases; JNK: c-Jun N-terminal kinases; MAPK: Mitogen-activated protein kinase; NF-κB: Nuclear factor-kappa B; NO: Nitric oxide; PG: Prostaglandin; PI3K: Phosphoinositide 3-kinase; PMN: Polymorphonuclear neutrophils; TNF: Tumor necrotic factor.

**Figure 6 molecules-24-03505-f006:**
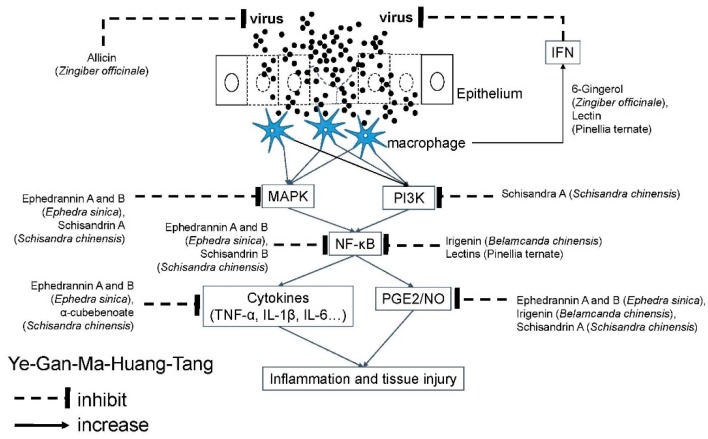
Molecular mechanism of Ye-Gan-Ma-Huang-Tang against airway viral infections. Airway viruses infect the epithelium to cause tissue injury. Ye-Gan-Ma-Huang-Tang contains several active molecules to inhibit viral replication and signal transduction of inflammatory response. Akt: Serine/threonine protein kinase B (PKB); COX: Cyclooxygenase; ERK: Extracellular signal-regulated kinase; IL: interleukin; iNOS: Inducible nitric oxide synthases; JNK: c-Jun N-terminal kinases; MAPK: Mitogen-activated protein kinase; NF-κB: Nuclear factor-kappa B; NO: Nitric oxide; PG: prostaglandin; PI3K: Phosphoinositide 3-kinase; PMN: Polymorphonuclear neutrophils; TNF: Tumor necrotic factor.

**Table 1 molecules-24-03505-t001:** Composition of Ge-Gen-Tang (GGT; Kakkon-to in Japan; Galgeun-tang in Korea; Ge-Gen-Tang in Chinese) [49].

Chinese Medicine Plant	Family	Weight (gm)	Used Part	Identified Molecules
*Cinnamomum cassia* (L.) J. Presl	Lauraceae	3.0	Twig	Coumarin, cinnamic alcohol, cinnamic acid, 2-methoxy cinnamic acid, cinnamaldehyde, E-cinnamaldehyde, 2-methoxy cinnamaldehyde, 6-methoxy cinnamaldehyde
*Ephedra sinica* Stapf.	Ephedraceae	4.5	Aerial part	Ephedrine, L-ephedrannin, ephedrannin A and B, pseudoephedrine alkaloid, flavonoids, and organic acids
*Glycyrrhiza uralensis* Fisch.	Leguminosae	3.0	Root and Stolon	Glycyrrhizin, glycyrrhinic acid, glycyrrhetic acic or glycyrrhetinic acid, liquiritin, liquiritigenin, glycyamarin, iso-liquiritin, grabric acid, licoricidin, glycyrol, 5-0-methyl glycerol, iso-glycyrol
*Paeonia lactiflora* Pall.	Ranunculaceae	3.0	Radix	Paeoniflorin, oxypaeoniflorin, albiflorin, benzoylpaeoniflorin, paeoniflorigenone, paeonolide, paeonol
*Pueraria lobata* Ohwi	Leguminosae	6.0	Radix	Puerarin, daidzin, genistin, daidzein, genistein
*Zingiber officinale* Roscoe	Zingiberaceae	4.5	Root-like stem	6-Gingerol, 6-Shogaol, zingerone, allicin
*Ziziphus jujuba* Mill.	Rhamnaceae	4.0	Fruit	3-O-(trans-p-coumaroyl)-alphitolic acid, 3-O-(cis-p-coumaroyl)-alphitolic acid, 3β-O-(trans-p-coumaroyl)-maslinic acid, pomonic acid, 2-oxo-pomolic acid, benthamic acid, terminic acid, oleanic acid, betulinic acid, quercetin 3-O-rutinoside, quercetin 3-O-robinobioside, apigenin, traumatic acid, (Z)-4-oxotetradec-5-enoic acid, 7(E)-9-keto-hexadec-7-enoic acid, 9(E)-11-oxo-octadecenoic acid (9CI), and magnoflorine

**Table 2 molecules-24-03505-t002:** Composition Ma-Huang-Tang (MHT; Maoto in Japan) [72].

Chinese Medicine Plant	Family	Weight (gm)	Used Part	Identified Molecules
*Ephedra sinica* Stapf.	Ephedraceae	9.0	Stalk	Ephedrine, L-ephedrannin, ephedrannin A and B, pseudoephedrine alkaloids, flavonoids, and organic acids
*Cinnamomum cassia* (L.) J. Presl	Lauraceae	6.0	Twig	Coumarin, cinnamic alcohol, cinnamic acid, 2-methoxy cinnamic acid, cinnamaldehyde, E-cinnamaldehyde, 2-methoxy cinnamaldehyde, 6-methoxy cinnamaldehyde
*Glycyrrhiza uralensis* Fisch.	Fabaceae	3.0	Root & Rhizome	Glycyrrhizin, glycyrrhinic acid, glycyrrhetic acic or glycyrrhetinic acid, liquiritin, liquiritigenin, glycyamarin, iso-liquiritin, grabric acid, licoricidin, glycyrol, 5-0-methyl glycerol, iso-glycyrol
*Prunus armeniaca* L. var. *ansu* Maxium	Rosaceae	5.0	Ripe seed	Amygdalin

**Table 3 molecules-24-03505-t003:** Composition Ma-Xing-Gan-Shi-Tang (MXGST) [88].

Chinese Medicine Plant	Family	Weight (gm)	Used Part	Identified Molecules
*Ephedra sinica* Stapf.	Ephedraceae	8.0	Stalk	Ephedrine, L-ephedrannin, ephedrannin A and B, pseudoephedrine alkaloids, flavonoids, and organic acids
*Prunus armeniaca* L. var. *ansu* Maxium	Rosaceae	6.0	Ripe seed	Amygdalin
*Glycyrrhiza uralensis* Fisch.	Leguminosae	4.0	Root and Rhizome	Glycyrrhizin, glycyrrhinic acid, glycyrrhetic acic or glycyrrhetinic acid, liquiritin, liquiritigenin, glycyamarin, iso-liquiritin, grabric acid, licoricidin, glycyrol, 5-0-methyl glycerol, iso-glycyrol
*Gypsum Fibrosum*	CaSO_4_·2H_2_O	16.0		

**Table 4 molecules-24-03505-t004:** Composition Xiao-Qing-Long-Tang (XQLT; Sho-seiryu-to in Japan; so-cheong-ryong-tang in Korea) [98].

Chinese Medicine Plant	Family	Weight (gm)	Used Part	Identified Molecules
*Ephedra sinica* Stapf	Ephedraceae	4.0	Stem	Ephedrine, L-ephedrannin, ephedrannin A and B, pseudoephedrine alkaloids, flavonoids, and organic acids
*Cinnamomum cassia* (L.) J. Presl	Lauraceae	4.0	Twig	Coumarin, cinnamic alcohol, cinnamic acid, 2-methoxy cinnamic acid, cinnamaldehyde, E-cinnamaldehyde, 2-methoxy cinnamaldehyde, 6-methoxy cinnamaldehyde
*Paeonia lactiflora* Pall.	Ranuculaceae	4.0	Root	Paeoniflorin, oxypaeoniflorin, albiflorin, benzoylpaeoniflorin, paeoniflorigenone, paeonolide, paeonol
*Glycyrrhiza uralensis* Fisch.	Leguminosae	4.0	Root	Glycyrrhizin, glycyrrhinic acid, glycyrrhetic acic or glycyrrhetinic acid, liquiritin, liquiritigenin, glycyamarin, iso-liquiritin, grabric acid, licoricidin, glycyrol, 5-0-methyl glycerol, iso-glycyrol
*Zingiber officinale* Roscoe	Zingiberaceae	4.0	Rhizome	6-Gingerol, 6-Shogaol, Zingerone, Allicin
*Pinellia ternata* (Thunb.) Breitenb.	Araceae	4.0	Tuber	3-Acetoamino-5-methylisooxazole, butyl-ethylene ether, 3-methyleicosane, hexadecylendioic acid, methyl-2-chloropropenoate, anethole, benzaldehyde, 1,5-pentadiol, 2-methylpyrazine, 9-heptadecanol, ethylpalmitate, pentaldehyde oxime, ephedrine, choline, β-ssitosterol, daucosterol, homogentisic acid, protocatechualdehyde, shogaol, baicaline, baicalein, gingerol, 1,2,3,4,6-penta-Ogalloylglucose, 12,13-epoxy-9-hydroxynonadeca-7,10-dienoic acid, aminobutyric acid, aspartic acid
*Asarum heterotropides* F.Schmidt. f*.mandshuricum* (Maxim.) Kitag.	Aristolochiaceae	1.5	Root	Methylleugenol, safraole, asatone, α- and β-pinene, asaricin, eucarvone, estragole
*Schisandra chinensis* (Turcz.) Baill	Magnoliaceae	1.5	Fruit	Deoxyschizandrin, γ-schizandrin, schizandrin, aomisin, pseudo-r-schizandrin, schisantherin A

**Table 5 molecules-24-03505-t005:** Composition Ye-Gan-Ma-Huang-Tang (YGMHT; Yakammaoto in Japan; Shegan-Mahuang-Tang or Sheganmahuang Decoction in Chinese) [115].

Chinese Medicine Plant	Family	Weight (gm)	Used Part	Main Identified Molecules
*Ephedra sinica* Stapf.	Ephedraceae	4.0	Stem	Ephedrine, L-ephedrannin, ephedrannin A and B, pseudoephedrine alkaloids, flavonoids, and organic acids
*Pinellia ternata* (Thunb.) Breitenb.	Araceae	4.0	Tuber	3-acetoamino-5-methylisooxazole, butyl-ethylene ether, 3-methyleicosane, hexadecylendioic acid, methyl-2-chloropropenoate, anethole, benzaldehyde, 1,5-pentadiol, 2-methylpyrazine, 9-heptadecanol, ethylpalmitate, pentaldehyde oxime, ephedrine, choline, β-ssitosterol, daucosterol, homogentisic acid, protocatechualdehyde, shogaol, baicaline, baicalein, gingerol, 1,2,3,4,6-penta-Ogalloylglucose, 12,13-epoxy-9-hydroxynonadeca-7,10-dienoic acid, aminobutyric acid, aspartic acid
*Zingiber officinale* Roscoe	Zingiberaceae	4.0	Rhizome	6-Gingerol, 6-Shogaol, zingerone, allicin
*Tussilago farfara L.*	Compositae	3.0	Flower	Faradiol, armiliot, rutin, hyperin, tussilagone, tannin, essential oil, wax
*Aster tataricus* L.f.	Compositae	3.0	Root & Rhizome	Shionone, epifriedelanol
*Ziziphus jujube* Mill.	Rhamnaceae	2.0	Fruit	3-O-(trans-p-coumaroyl)-alphitolic acid, 3-O-(cis-p-coumaroyl)-alphitolic acid, 3β-O-(trans-p-coumaroyl)-maslinic acid, pomonic acid, 2-oxo-pomolic acid, benthamic acid, terminic acid, oleanic acid,betulinic acid, quercetin 3-O-rutinoside,quercetin 3-O-robinobioside, apigenin,traumatic acid, (Z)-4-oxotetradec-5-enoic acid, 7(E)-9-keto-hexadec-7-enoic acid, 9(E)-11-oxo-octadecenoic acid (9CI), magnoflorine
*Belamcanda chinensis* (L.) DC.	Iridaceae	1.5	Rhizome	Irisflorentin, isorhapontigenin, tectorigenin
*Asarum heterotropides* F.Schmidt. f*.mandshuricum* (Maxim.) Kitag.	Aristolochiaceae	1.5	Root	Methylleugenol, safraole, asatone, α- and β-pinene, asaricin, eucarvone, estragole
*Schisandra chinensis* (Turcz.) Baill	Magnoliaceae	1..0	Fruit	Deoxyschizandrin, γ-schizandrin, schizandrin, aomisin, pseudo-γ-schizandrin, schisantherin A

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
