# Peer review of "Unraveling the Molecular Mechanism of Traditional Chinese Medicine: Formulas Against Acute Airway Viral Infections as Examples"

_molecules, 2019, doi:10.3390/molecules24193505_

Round 1

Reviewer 1 Report

It is a very comprehensive review paper that explains all kinds of interactions that may be induced by herbal compounds/extracts. Some interesting examples of herbal formulae used for viral infections are discussed . 

Several suggestions:

It looks like a chapter in a thesis rather than a paper. It has too much information and does not have a stand-out story.  Introduction of TCM is too long, section 1-3 is not directly linked to management of viral infections! Therefore, your content does not fully reflect the title but more like an overview of complex molecular mechanisms and possible interactions of TCM. Section 5 discusses the limitation of TCM, however, how does that link with the title - mechanistic studies? Please check the grammar and font of text thoroughly  It would be good to add a column of associated pathways to table 1- 5, or even have a signalling pathway maps (similar to Figure 1). Then it is very clear of how those compounds interact with each other or with their targeted receptors.  Most of the bioactive compounds in the herb have very low amount in the herb and formula. Therefore, the likelihood of interaction is even lower if it is administered as in a decoction or formula. You may discuss a bit more of the clinical significance.  

Author Response

It looks like a chapter in a thesis rather than a paper. It has too much information and does not have a stand-out story. Answer: We would like to thank reviewer for this kind reminding. We completely agree that this review provide much information and does not have a stand-out story. However, the purpose of this review is to summarize the current knowledge and shortage of molecular mechanisms of traditional Chinese medicine (TCM) using TCM formulas against airway viral infection for examples. Therefore, we slightly modified the title into "Unraveling The Molecular Mechanism of Traditional Chinese Medicine: formulas Against Acute Airway Viral Infections as Examples". We suppose that our work will contribute to the aim of this supplement. Thank you very much. Introduction of TCM is too long, section 1-3 is not directly linked to management of viral infections! Therefore, your content does not fully reflect the title but more like an overview of complex molecular mechanisms and possible interactions of TCM. Section 5 discusses the limitation of TCM, however, how does that link with the title-mechanistic studies? Answer: We would like to thank reviewer for this kind suggestion and reminding. We completely agree that section 1-3 are not directly linked to the management of viral infections that does not fully reflect the title. We also agree that section 5 cannot link with the title- mechanistic studies. To make the content fully reflecting the title, we slightly modified the title. We hope that our revision is satisfactory. Thank you very much. Please check the grammar and font of text thoroughly. Answer: Thank you very much for this kind suggestion. We have asked a native English speaker to check the grammar and font of text. We are glad to provide you the revision and hope that our revision is satisfactory. Thank you very much. It would be good to add a column of associated pathways to table 1-5, or even have a signalling pathway maps (similar to Figure 1). Then it is very clear of how those compounds interact with each other or with their targeted receptors. Answer: We would like to than reviewer for this kind suggestions. We have supplemented figure of signaling pathway to each TCM formula to make it clearer for readers to understand their mechanisms. Most of the bioactive compounds in the herb have very low amount in the herb and formula. Therefore, the likelihood of interaction is even lower if it is administered as in a decoction or formula. You may discuss a bit more of the clinical significance. Answer: We would like to thank reviewer for this kind correction and comment. We completely agree that most of the bioactive compounds in the herb and formulas have very low amount. However, with a little amount fo bioactive compounds, herbs and TCM formulas actually have bioactivities, including clinical effects and side effects. This is why we need to unravel their molecular mechanisms. We have supplemented the discussion in section 3 for overview, in section 5.2 for effectiveness, and in section 5.4 for interactions. (Please see the attachment) We would like to thank reviewer gratefully for these kind reminding and suggestions. We hope that our answers are satisfactory. Thank you very much.

Reviewer 2 Report

Generally speaking this is an interesting and potentially valuable paper, especially if observed from the aspects of integrative medicine. However, to fulfill such expectations two chapters should be added:

positioning TCM within modern concepts of integrative medicine, preferably addressing pathophysiology of oxidative stress describing relationships between herbal and non-herbal TCM treatments such as acupuncture

Author Response

Response to the reviewer 2:

Generally speaking, this is an  interesting and potentially valuable paper, especially if observed from the aspects of integrative medicine. However, to fulfill such expectations two chapters should be added: positioning TCM within modern concepts of integrative medicine, preferably addressing pathophysiology of oxidative stress describing relationships between herbal and non-herbal TCM treatments such as acupuncture. Answer: We would like to thank reviewer for this valuable suggestion. We completely agree that positioning TCM within modern concepts of integrative medicine need to address oxidative stress in relation to herbal and non-herbal TCM treatments such as acupuncture. We have stated in section 2 "TCM includes herbal therapy, acupuncture, massage, and dietary therapy. In the current work, TCM will be simply defined as the herbal and dietary therapies". Acupuncture is and important part of TCM. Its effect involves anti-oxidation pathway, neuroprotective, and anti-inflammatory processes [1]. Acupuncture has been proved to decrease inflammatory factors, such as TNF-alfa, IL-1beta, IL-6, COX-1, COX-2, and PGE2 expression [2-4]. However, the current work focuses on the herbal and dietary therapies. Therefore, we did not include acupuncture. As for oxidative stress, it is quite important to clarify herbal treatment in relation to oxidative stress. Therefore, we have supplemented a section 5.6 Pitfall of interpretation of benefits. Please see the attachment. 

   We would like to thank reviewer gratefully for the kind reminding and suggestions. We hope that our answers are satisfactory. Thank you very much.

Round 2

Reviewer 1 Report

Revisions have been done accordingly, and I have no further comments.